# Iron Metabolism in the Tumor Microenvironment—Implications for Anti-Cancer Immune Response

**DOI:** 10.3390/cells10020303

**Published:** 2021-02-02

**Authors:** Alessandro Sacco, Anna Martina Battaglia, Cirino Botta, Ilenia Aversa, Serafina Mancuso, Francesco Costanzo, Flavia Biamonte

**Affiliations:** 1Department of Experimental and Clinical Medicine, “Magna Graecia” University of Catanzaro, 88100 Catanzaro, Italy; alessandro.sacco@studenti.unicz.it (A.S.); annamartinabattaglia@gmail.com (A.M.B.); ilenia.aversa@unicz.it (I.A.); fsc@unicz.it (F.C.); 2Annunziata Hospital, 87100 Cosenza, Italy; cirino.botta@gmail.com; 3U.O. Biochimica Clinica, Azienda Ospedaliero Universitaria Mater Domini, 88100 Catanzaro, Italy; serafina.mancuso@materdominiaou.it; 4Center of Interdepartmental Services (CIS), “Magna Graecia” University of Catanzaro, 88100 Catanzaro, Italy

**Keywords:** iron metabolism, ferroptosis, cancer, innate immune response, adaptive immune response tumor microenvironment

## Abstract

New insights into the field of iron metabolism within the tumor microenvironment have been uncovered in recent years. Iron promotes the production of reactive oxygen species, which may either trigger ferroptosis cell death or contribute to malignant transformation. Once transformed, cancer cells divert tumor-infiltrating immune cells to satisfy their iron demand, thus affecting the tumor immunosurveillance. In this review, we highlight how the bioavailability of this metal shapes complex metabolic pathways within the tumor microenvironment and how this affects both tumor-associated macrophages and tumor-infiltrating lymphocytes functions. Furthermore, we discuss the potentials as well as the current clinical controversies surrounding the use of iron metabolism as a target for new anticancer treatments in two opposed conditions: (i) the “hot” tumors, which are usually enriched in immune cells infiltration and are extremely rich in iron availability within the microenvironment, and (ii) the “cold” tumors, which are often very poor in immune cells, mainly due to immune exclusion.

## 1. Introduction

In recent years, the role played by the tumor microenvironment (TME) in fostering or preventing tumor growth has gained considerable attention. Cellular and non-cellular components in TME may reprogram tumor initiation, expansion, and progression, thus serving as potential targets for cancer therapy [1,2,3]. Cancer research as well as cancer therapies have indeed switched from a “cancer-centric” model to a “TME-centric” one [4].

According to the new concept proposed by Ming-Zhu Jin and Wei-Lin Jin, the TME includes diverse specialized microenvironments that overlap and cooperate with one another [4]. Among them, immune and metabolism microenvironments mutually influence their functionality to establish a pro-tumoral and immunosuppressive environment or, otherwise, to support the anti-tumoral immune response [5,6,7].

Tumor immune microenvironment (TIME) is broadly populated with tumor-infiltrating immune cells, such as B and T lymphocytes, natural killer (NK) cells, tumor-associated macrophages (TAMs), and myeloid-derived suppressor cells (MDSCs) [8].

The compartment of TAMs is highly dynamic and heterogeneous, and accounts for up to 50% of the tumor’s bulk in most solid as well as in several hematologic malignancies [8,9,10,11,12]. TAMs can acquire diverse phenotypic, metabolic, and functional profiles, hovering between a pro-inflammatory (so-called M1) to an alternatively anti-inflammatory (so-called M2) state [13,14], although this dichotomy is now considered quite an oversimplification. Born to mirror the classic T helper 1 (T_h_1) and T helper 2 (T_h_2) dualism, macrophage classification suffers from different limitations that should be carefully taken into account when using it: (1) It reflects a response to in vitro stimuli that are not present “alone” within tumor tissue: Macrophages usually derive from monocytes differentiation whose transcriptional activation depends upon the context and the number of stimuli received, often giving rise to mixed phenotypes; (2) macrophages are very plastic cells: Their functional phenotype could be modified from their interaction with other immune cells, cancer cells or pathogens; (3) in contrast to T cells, macrophages do not expand clonally, thus each macrophage could be different from another one [15,16,17]. Nevertheless, it is currently believed that while specific TAM subsets support tumor initiation and progression, other TAM populations restrain tumor progression [18,19]. Within this review, we will use the terms “M1-like” and “M2-like” TAMs to indicate to which one of the classical phenotypes TAMs are close in each scenario, considering the impossibility to obtain a clear specific phenotype.

Among the adaptive immune system, CD4^+^ and CD8^+^ T_eff_ cells regulate and fulfil the antigen-specific killing of cancer cells, respectively. Particularly, CD8^+^ T_eff_ cells directly kill tumor cells by promoting apoptosis and by releasing a specific subset of cytokines. CD4^+^ T cells include numerous subgroups. The T_h_1 subset exerts an antitumor function through direct tumor cell killing or cytokine secretion, but also by assisting in CD8^+^ T cell activation. On the other hand, CD4^+^ regulatory T (Treg) cells show an immune suppressive and a pro-tumorigenic activity [20,21,22,23,24].

Cancer-associated fibroblasts (CAFs), representing a high proportion of non-cancer cells in the TME, also support tumor expansion and invasiveness by remodeling the surrounding stroma. Iron mediated production of reactive oxygen species (ROS) promotes the activation state (myofibroblast-like) of CAFs which, in turns, release growth factors, extracellular matrix (ECM) remodeling factors and cytokines, thus promoting tumor metastasis [25]. On the other hand, CAFs support iron utilization of breast cancer cells by inducing the expression of the iron-regulatory hormone hepcidin [26].

It is now well established that metabolic reprogramming in immune cells tightly correlates with their phenotype and functions [27]. Moreover, highly active metabolic pathways within cancer cells can affect the composition of nutrients within the TME, thus impairing the immune response [27,28,29,30,31,32,33].

Compared to normal cells, tumor cells exhibit iron addiction [34,35,36]. In particular, in metastatic cancers, iron accumulation is associated with the epigenetic regulation of genes involved in the epithelial to mesenchymal transition (EMT) and in the development of cancer stem cells (CSCs) features [37,38]. Cancer cells show an iron-sequestering phenotype achieved through the activation of iron uptake processes and the parallel downregulation of the export pathways. In addition, tumor cells also divert immune cells resident in TME to satisfy their enhanced demand of iron supply [39].

Iron is able to promote tumor cell growth by acting as a cofactor of enzymes involved in ATP production (i.e., cytochrome-c reductase), antioxidant defense (i.e., superoxide dismutase, SOD), DNA replication, and repair (i.e., DNA polymerases) [40,41,42,43,44,45,46]; furthermore, it can act as a metal catalyst of demethylases enzymes involved in the epigenetic plasticity underlying EMT and cancer progression [38]. In this regard, it has been recently demonstrated that in persister cancer cells, intracellular iron acts as metal catalyst of α-ketoglutaric acid (αKG)-dependent demethylases, which repress histones methylation and promote the expression of mesenchymal genes [38]. Iron, however, can participate to potentially deleterious ROS-generating reactions, which alter the redox balance of the cell and potentially cause oxidative damage, thus ultimately leading to an iron-dependent programmed cell death called ferroptosis [47,48,49,50,51]. However, the progression of cancer cells leads to the development of tolerance to ROS accumulation and to a new redox homeostasis, which may enhance tumorigenicity [4].

To adequately review this topic, in this review, we address the following key questions: i. How iron handling by TAMs affects tumor cell growth, ii. how iron can regulate the adaptive immune response, iii. how ferroptosis impacts tumor immunity, and iv. how iron modulation can be employed in combination with existing immunotherapies to enhance their efficacy.

## 2. Intracellular Iron Metabolism

Circulating ferric iron (Fe^3+^) is bound by transferrin (TF) and then delivered to cells and tissues. The transferrin receptor (TFR1, CD71), expressed on cell surface, internalizes Fe^3+^-loaded TF through an endocytosis-mediated mechanism. Thus, ferric iron is released and TFR1 is recycled to the cell surface. The ferrireductases STEAP2/3/4 convert Fe^3+^ into its ferrous form Fe^2+^ [52,53], which, in turn, enters the cytoplasm via divalent metal transporter 1 (DMT1). Non-transferrin bound iron (NTBI) enters the cytoplasm through different carrier molecules, such as DMT1, the zinc transporters ZIP8/14, and the L-type voltage-dependent calcium channels [54]. Notably, recent evidence indicate that EMT enhances CD44/hyaluronate-mediated iron endocytosis as an alternative iron-uptake process, preferentially involving the persister CSCs [38].

Fe^2+^, the free and redox-active iron, enriches the labile iron pool (LIP), promoting ROS generation through the Fenton reactions. Ferritin (FT), the main iron-storage protein composed of 24 subunits of heavy (FtH) and light (FtL) chains, prevents ROS formation by sequestering Fe^2+^, which is coupled to oxidation to Fe^3+^ [45]. When the intracellular levels are low, iron is released from FT nanocage through several iron-reductive mobilization pathways mediated by different biomolecules, such as flavin nucleotides and glutathione [55,56]. Besides, a process called ferritinophagy has been described. This event is mediated by the nuclear receptor coactivator 4 (NCOA4), which binds FT triggering its autophagic degradation [57]. Ferritin levels are also affected by oxidative stress and inflammation through the activation of the transcription factors NF-kB and NRF2, as well as through the release of proinflammatory cytokines, such as IL-6 and IL-1. NF-kB and NRF2 increase FT transcription, while IL-6 and IL-1 stimulate FT translation, thus ensuring Fe^2+^ storage [58,59,60]. In addition, the metallochaperones poly(C)-binding proteins 1/2 (PCBP1/2) bind and facilitate iron loading into ferritin [61]. Most of the LIP, however, is internalized into mitochondria through the mitoferrin transporters (Mfrn1/2) [51]. Mfrn1/2 import Fe^2+^ from the intermembrane space to the mitochondrial matrix, where it is used for heme and iron-sulfur (Fe-S) cluster biogenesis mediated by frataxin and GLRX5 enzymes [62,63]. Mitochondria are also provided by a specific H-type of ferritin (FtMt) devoted to mitochondrial iron storage [64].

Intracellular iron export is mediated by ferroportin (FPN) [65]. FPN exports Fe^2+^ to the extracellular space where it is oxidized by ceruloplasmin (CP) [66], hephaestin (HEPH) [67], and zyklopen (HEPHL1) [66]. FPN is expressed in numerous cell types, in particular, in those involved in the regulation of plasma iron levels, such as enterocytes, macrophages, and hepatocytes [68]. FPN activity is decreased by the liver-derived hormone hepcidin. Recent studies highlight that hepcidin degrades FPN loaded with iron by binding to a specific cavity located between the N and C domains, thus blocking iron efflux and inhibiting its transport [69]. This results in an overall reduction of iron in the bloodstream [70].

The proteins involved in intracellular iron metabolism are summarized in Table 1.

## 3. Iron Handling by TAMs and Its Implication for Cancer Progression

Tissue-resident macrophages are the “gate-keepers” of iron homeostasis. Indeed, they take up iron, metabolize it, store it, and export it to satisfy the requests of the surrounding cells [71,72,73]. Intracellular iron metabolism could move the balance of polarization in the direction of the “classically activated” M1- and the “alternatively activated” M2-macrophages according to diverse microenvironmental stimuli and to local metabolic cues [74,75]. M1-macrophages are activated by T_h_1 cell-derived interferon-γ (IFN-γ) and pathogen-associated molecular patterns (PAMPs) signaling as well as by the interaction with Toll-like receptors (TLRs). M1-macrophages produce pro-inflammatory mediators such as IL-6, IL-1β, IL-12, and tumor necrosis factor-α (TNF-α) as well as ROS and nitrogen species. Conversely, M2-macrophages are stimulated by T_h_2 cells-derived IL-4, IL-13, and by IL-10, and are responsible for the release of an alternative repertoire of cytokines that help to resolve inflammation [76,77]. Considering their functional diversity, it is not surprising that macrophages show distinct properties in managing iron metabolism [78].

Similar to tissue-resident macrophages, TAMs are characterized by an elevated plasticity. Indeed, in relation to the heterogenous conditions they are exposed to within the TME, TAMs can acquire different phenotypic and functional profiles, ranging from an M1-like to an M2-like state [79,80]. TAMs heterogeneity has been revealed both within and across tumors [10,12,81]. While specific TAM subsets support tumor initiation, progression, and immunoevasion, other TAM populations exert anti-tumoral activity, thus sustaining the efficacy of several immunotherapies [82,83].

In most types of tumors, TAMs display an M2-polarized phenotype. M2-like TAMs, preferentially homing to hypoxic and necrotic areas of the TME, present an iron-release prone phenotype, characterized by high levels of FPN and low levels of FT, respectively [39]. These features allow the M2-like TAMs to promote iron recirculation in the TME and to support tumor cell proliferation, angiogenesis, and metastasis [84,85,86]. Notably, FT^low^, FPN^high^ TAMs increase the expression of CD91 or CD163 to specifically uptake hemopexin-heme or haptoglobin-hemoglobin as other important sources of iron [87,88]. Furthermore, through the increased phagocytosis of senescent erythrocytes into erythrophagosomes, M2-like TAMs promote intracellular heme accumulation [89]. Once internalized, heme is then degraded by Heme oxygenase (HMOX-1) into its metabolic products biliverdin, carbon monoxide, and Fe^2+^. The latter inhibits iron binding proteins 1/2 (IRP1/2) interaction to *FPN* mRNA, thus promoting its translation [90]. Otherwise, intracellular heme accumulation promotes the release of transcriptional repressor Bach-1 from the specific target sequences within *FPN* promoter region, thus enabling its transcription [84].

Data obtained in human breast cancer support the existence of alternative FPN-independent iron transport routes in the TME. Two reports highlight a critical role of macrophage-secreted lipocalin-2 (LCN-2), a protein able to bind siderophore-complexed iron and to export it into the TME, in the promotion of cancer cells proliferation in vitro [91,92]. In the TAMs, LCN2 colocalizes with the iron-binding glycoprotein lactoferrin and promotes the release of pro-inflammatory cytokines into the TME [93]. LCN2 can also be released into the extracellular matrix where it binds to MMP-9, resulting in matrix degradation and EMT [92,94].

Iron export in TAMs is also mediated by the secretion of FT, which acts as a tumor growth factor to promote the proliferation of breast cancer cells [95], regardless of its iron content. An opposite role of this protein in the anti-tumor immunity has been described as well [96,97,98,99].

In cancer associated with chronic inflammation, TAMs show an M1-like phenotype or overlapping M1/M2 features [100,101]. M1-like TAMs foster iron uptake and storage and display a mitigated iron-release phenotype. Indeed, they show a marked iron absorbing activity either through TFR1 or NTBI transporters, such as ZIP8/14 [102,103]. M1-like TAMs express high levels of FT, whereas the FPN is less expressed, causing intracellular iron retention [39,104]. A recent study highlighted that, upon iron overload, macrophages acquire a pro-inflammatory phenotype and that this is associated with ROS generation, enhanced p300/CBP acetyltransferase activity, and increased p53 acetylation [105]. Iron-loaded TAMs infiltration correlates with tumor regression in NSCLC patients, suggesting that targeted iron delivery to TAMs can be used as adjuvant therapeutic strategy to improve antitumor immune response [106]. All these mechanisms are summarized in Figure 1.

## 4. The Role of Iron in the Control of T Lymphocytes Functions

A sufficient iron supply is required for the activity of many heme- and Fe-S-containing enzymes involved in the ATP-generating metabolic reactions, as well as in cell division, essential for T cell growth, expansion, and functions [107]. One of the earliest events of T cell activation and proliferation is the upregulation of TFR1 [108,109,110]. In agreement, mutations in the gene encoding TFR1 impair T cell function [111]. Upon intake, intracellular iron is then stored in FT whose activity is required for T cell proliferation [112,113,114,115]. Indeed, genetic deletion of the heavy subunit of FT, FtH, causes an increase of the intracellular labile iron pool and ROS production, which, in turn, break down T cell expansion [116]. In triple-negative breast cancer, the subcellular localization of FtH affects its own role in the immune response. Indeed, cytoplasmic FtH in breast cancer cells regulates the MHC-I-mediated antigen processing and presentation, thus consequently inducing the recall of CD8^+^ T cells, whereas nuclear FtH promotes cancer cells viability [112]. In patients with melanoma, serum FtH is associated with enhanced circulating T_reg_ cells and supports their immune functions [117].

Overall, these data suggest that an optimal homeostasis of iron metabolism is crucial for T cell function.

## 5. Ferroptosis Enhances Antitumor Immunity

Ferroptosis is a programmed cell death caused by the iron-dependent accumulation of ROS to toxic levels [51,118]. So far, several signaling pathways underlying ferroptosis have attracted attention in cancer research: (i) The increase of intracellular iron, which elevates ROS levels and leads to intense membrane lipid peroxidation, (ii) the inactivation of the antioxidant glutathione-dependent peroxidase 4 (GPX4), a selenoprotein required for an efficient reduction in peroxidized phospholipids, (iii) the repression of the cystine-glutamate antiporter (system Xc-) [119], which hinders the cellular influx of cystine thus inhibiting glutathione (GSH) synthesis and further preventing GPX4 activity [120,121,122]. Recent studies also defined ferroptosis as an autophagic type of cell death [123]. Indeed, by degrading ferritin via ferritinophagy, NCOA4 promotes the increase of LIP and the following ROS-generating Fenton reactions in fibroblasts and cancer cells [57].

Lately, ferroptosis has been linked to antitumor immunity and potentially included in the concept of immunogenic cell death (ICD) [124]. ICD is defined by the chronic release or membrane exposure of damage-associated molecular patterns (DAMPs), which act as danger signals to recruit and activate several immune cells in the TME [125]. The induction of ICD is dependent on stress stimuli, since endoplasmic reticulum (ER) stress and ROS production are needed for the exposure of different DAMPs (i.e., ATP, HMGB1) [126].

The first evidence of the connection between ferroptosis and antitumor immunity has been provided by Wang et al. These authors show that immunotherapy-activated CD8+ T lymphocytes induce ferroptosis in cancer cells through the downregulation of *SLC7A11* and *SLC3A2* genes encoding for the two subunits of system Xc-. Mechanistically, Wang et al. demonstrate that the tumor cell coculture with IFN-γ-rich supernatant obtained from activated T cells induces lipid ROS generation and ferroptosis. The molecular bases underlying this phenomenon are most likely associated to the IFN-γ-induced transcriptional inhibition of *SLC7A11* and *SLC3A2* genes mediated by STAT1. Indeed, in tumor cells lacking STAT1, IFN-γ is unable to downregulate *SLC7A11* as well as to foster RSL3-induced lipid peroxidation and cell death. In contrast, tumor cells treated with the ferroptosis inhibitor liproxstatin-1 are insensitive to the anti-PD-L1 therapy [127]. Interestingly, in melanoma patients that benefit from immunotherapy, a higher T cell infiltration, as well as an increased release of IFN-γ, correlate with low expression of *SLC3A2* gene in the relative cancer cells [127].

Furthermore, the same team demonstrates that ferroptosis can be triggered by the combined action of IFN-γ and ataxia-telangiectasia mutated (ATM) activated by radiotherapy in melanoma cells and human fibrosarcoma cells [128].

Recent experimental data prove that cancer cells undergoing ferroptosis release High mobility group Box 1 (HMGB1) in an autophagy-dependent manner [129,130]. Upon cancer cell death, HMGB1 is released in the surrounding TME where it physically interacts with several pattern recognition receptors (PRRs), such as TLR2, TLR4, and RAGE, thus stimulating the innate immune system [131,132]. In particular, it has been demonstrated that either chemotherapy or radiotherapy lead to HMGB1 release from dying cancer cells, promoting antigen processing and presentation on dendritic cells (DCs) through TLR4-MyD88 axis [133,134].

ROS-mediated ferroptosis promotes the translocation of calreticulin (CRT), a soluble ER-associated chaperone, on the surface of tumor cells [135,136]. In stressed or dying cells, CRT is exposed on the plasma membrane where it functions as a potent “eat-me” signal [137]. The prophagocytic “eat me” CRT signal induces robust antitumor immune responses by eliciting phagocytosis of tumor-associated antigens [138,139,140,141].

Finally, based on accumulating evidence, during ferroptosis tumor cells provide arachidonic acid (AA) for the biosynthesis of eicosanoids, which have been proved to promote antitumor immunity [142]. Otherwise, the induction of ferroptosis in tumor cells can be also associated with the release of prostaglandin E2 (PGE2) [120], which facilitates tumor evasion of immune surveillance [143,144]. Therefore, the production of PGE2 may represent an intrinsic impediment to the induction of a robust immune response by ferroptotic cells.

The crosstalk between ferroptosis and anticancer immunity is summarized in Figure 2.

## 6. Iron in TIME as New Target for Oncotherapy

Based on these premises, it is conceivable to hypothesize iron metabolism as a potential target to enhance the anti-cancer activity of current therapies. Specifically, we could face two different extreme conditions (with the great majority of cancers falling between these two: hyperinflamed tumors, which are usually enriched in immune cells infiltration and are extremely rich in iron availability within the microenvironment; and “cold” tumors, which are often very poor in immune cells, mainly due to immune exclusion (Figure 3).

In hyperinflamed tumors, as previously reported, the microenvironment is usually enriched in iron, released from inflammatory cells (TAM and TAN), which support cancer cell progression by inducing T and B lymphocytes anergy and death (due to oxidative stress [145]) and by reducing the capability of antigen-presenting cells (APCs) to elicit an effective anti-tumor immune response [146]. It is therefore clear that, in these conditions, an iron chelation therapeutic strategy could potentially be of utmost relevance to trigger (or release) an efficient anti-tumor response in cancer patients. Indeed, iron chelating agents such as deferasirox, an oral iron chelator currently used for the treatment of iron overload, demonstrated the capability to increase the Th1 response and to increase CD8 lymphocyte count in animal models of infections [147]. Additionally, by reducing regulatory T cells and enhancing the NK response, deferasirox improved the outcome of patients affected by acute myeloid leukemia after allogeneic stem cell transplantation (i.e., promoted a graft-versus-leukemia response) [148]. Interestingly, most of these agents, including deferasirox, ciclopirox olamine, desferrioxamine, and triapine, demonstrated direct anti-cancer properties and are currently under deep investigation as “companions” of standard chemotherapeutic schedules [146]. Furthermore, natural compounds with known anti-inflammatory and tumoricidal properties, including green tea, silybin, and curcumin recently demonstrated to act as iron chelators, opening new avenues for their use as “adjuvant” therapies for selected cancers treatment [129,146].

On the other hand, it should be recognized that cancer cells and CSCs are extremely susceptible to ferroptosis [149]. Along this line, different therapeutic strategies resulting in LIP increase could easily saturate the (limited) antioxidant cancer cell defenses and trigger ferroptosis. Indeed, radiotherapy and drugs such as ferumoxytol (iron nanoparticles approved for the treatment of iron deficiency) or erastin (and its analogues) are currently under active investigation in early clinical trials as combination partners for standard regimens due to their capability of acting as ferroptosis inducers [150]. Furthermore, according to Mai T.T. et al., treatment with salinomycin and its derivative ironomycin (AM5) selectively induces ferroptosis in breast CSCs by promoting iron sequestration in lysosomes. As demonstrated both in vitro and in vivo, these compounds, although leading to cytoplasmic iron depletion, do not act as regular iron chelators, since they promote ferritin degradation in lysosomes, further lysosomal iron accumulation, and the consequent lysosomal membrane permeabilization [37]. Overall, these findings suggest that lysosomal iron can be considered a promising druggable target.

However, the “double edge sword” role of ferroptosis in antitumor immunity should be carefully taken into account [151]. Indeed, while ferroptotic cancer cells could secrete a number of danger signals (AA metabolites and HMGB1) that foster anti-tumor immunity through lymphocyte recruitment and APC activation, both cancer and tumor-infiltrating immune cells (which could eventually undergo ferroptosis) could (at the same time) release immunosuppressive molecules such as PGE2 or overexpress immune checkpoints ligands to promote immune-escape [151]. Therefore, the overall result is strongly dependent on the tumor-associated microenvironment, thus making “cold” tumors the potential ideal candidates for triggering ferroptosis in cancer cells, bringing all the benefits of an “immunogenic cell-death” without the drawback of immune-suppression (“cold” tumors are virtually immune-excluded).

## 7. Discussion

The functional, metabolic, and immunological features of the TME are heterogeneous and dynamically evolve in response to both cell-intrinsic and cell-extrinsic factors [152,153]. As discussed above, the activation status and the specific functions of immune cells rely on major shifts in iron metabolism [154]. M2 TAMs, which are “iron-releasing”, appear to sustain cancer growth while “iron-retaining” M1 (potentially) limit tumor progression [155]. Consequently, targeting iron homeostasis in immune cells, and in particular macrophages, has received recent interest. However, whether it is better to inhibit iron utilization or to refuel TME with iron still remains an open question.

Iron-loaded macrophages show a pro-inflammatory M1-like phenotype, which may be used to induce anti-cancer responses. Indeed, in lung cancer, iron-loaded TAMs enhance the generation of ROS and pro-inflammatory cytokines (TNFα and IL-6), thus inducing tumor cell death [106]. However, the extreme plasticity of this compartment, characterized by quick and complete TME-related M1 to M2 phenotypic shifts (and vice versa) [156], renders this area of strong interest for future investigation.

Furthermore, new evidence suggests that immune checkpoint blockade in animal models reduces tumor growth in a ferroptosis-dependent manner [124], thus prompting the exciting possibility of delivering iron into TME to promote ferroptosis as an adjuvant therapeutic strategy to improve clinical benefits. On the other hand, in hyperinflamed tumors, the microenvironment is already (over-)enriched in iron, which, in turn, promotes cancer progression and immune evasion (T cell dysfunction). In line with this model, a therapeutic strategy based on iron chelation could potentially trigger an efficient anti-tumor response in cancer patients.

Apparently, limitations and toxicity concerns for each approach still need to be overcome to make iron targeting an effective therapeutic strategy. Along this line, the development of innovative tools for the identification of biomarkers of iron metabolism within the TME is eagerly awaited. Ultimately, the identification and optimization of possible combinatorial strategies that take into account the iron levels within the TME will surely improve the outcomes of patients with advanced cancer.

## Figures and Tables

**Figure 1 cells-10-00303-f001:**
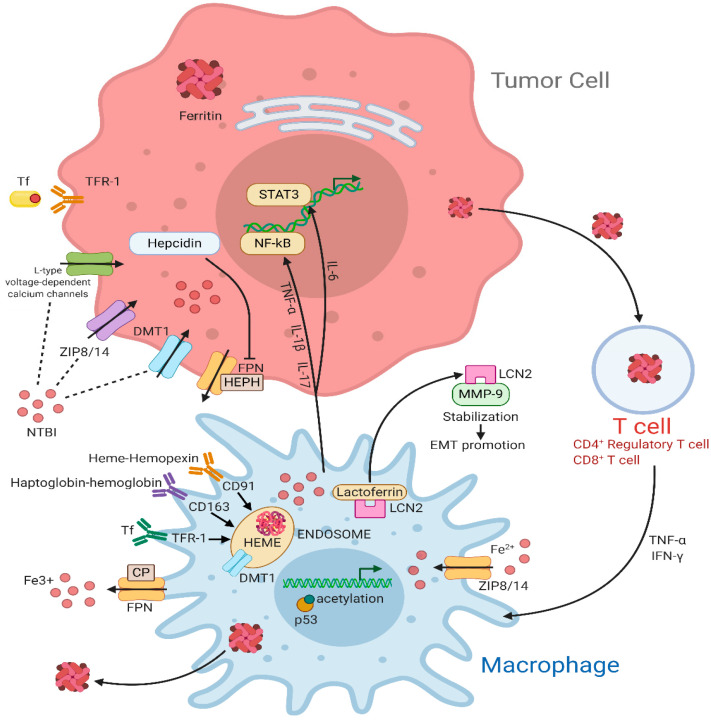
Iron cycle in the tumor microenvironment (TME). In the tumor cell, TFR1 internalizes Fe^3+^-loaded TF through an endocytosis-mediated mechanism. In addition, NTBI uptake is mediated by DMT1, ZIP8/14, and the L-type voltage-dependent calcium channels. When not used or stored, excess iron can be exported through FPN, an iron efflux pump coupled with HEPH or CP, two proteins with ferroxidase activity, to maintain iron homeostasis. FPN activity is decreased by hepcidin that directly binds to it. Inflammatory cytokines such as TNF-α, IL-1β, and IL-17 upregulate the NF-κB pathway while IL-6 acts on STAT3 pathway. Macrophages are major actors for iron metabolism, exportation, and storage in the tumor microenvironment. They can supply iron to support tumor growth by multiple transport pathways. Activated macrophages sequester iron through TF via TFR1 or through NTBI via ZIP8/14 transporters. Furthermore, hemopexin-heme and haptoglobin-hemoglobin, whose uptake is mediated by the interaction with CD91 and CD163, respectively, are consumed into endosomes. The iron-binding glycoprotein Lactoferrin colocalizes with LCN2, which, in turn, sequesters iron in the extracellular space and stabilize MMP-, thus promoting cell survival and EMT. FtH can be accumulated in circulating T cells preserving immune functions. Moreover, T cells can secret cytokines like TNF-α and IFN-γ, which increase DMT1 and decrease FPN level, resulting in increased iron retention. Abbreviations used: TFR-1, transferrin receptor; TF, transferrin; DMT1, divalent Metal (Ion) transporter 1; FPN, ferroportin; HEPH, hephaestin; CP, ceruloplasmin; TNF- α, tumor necrosis factor-α; IL-1β, interleukin-1β; IL-17, interleukin-17; LCN2, lipocalin 2; NF-κB, nuclear factor kappa-light-chain-enhancer of activated B cells; IL-6, interleukin-6; STAT3, signal transducer and activator of transcription 3; NTBI, non-transferrin bound iron; ZIP8/14, ZRT/IRT-like protein 8/14; CD91, cluster of differentiation 91; CD163, cluster of differentiation 163; MMP-9, matrix metalloproteinases-9; EMT, epithelial-mesenchymal transition; IFN-γ, interferon-γ.

**Figure 2 cells-10-00303-f002:**
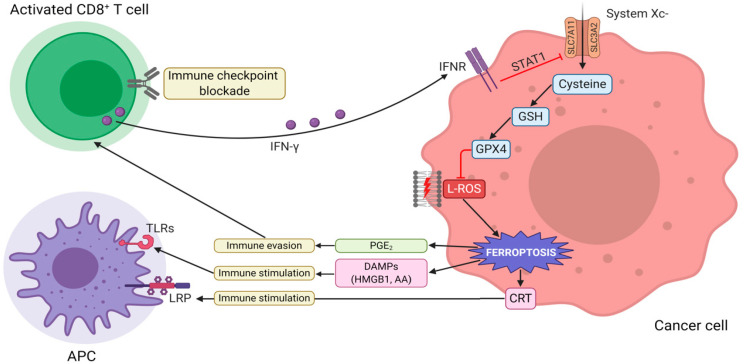
The interaction between tumor immunity modulation and ferroptosis. Immunotherapy-activated CD8+ T cells release IFN-γ, activating INFR. INFR inhibits SLC7A11 transcription through STAT1. This event leads to the downregulation of system Xc-, thus, triggering L-ROS-mediated ferroptosis via GSH-GPX4 axis. Ferroptotic cells release DAMPs (such as HMGB1, CRT, and AA), which, in turn, promote the recruitment and activation of immune cells. Vice versa, the increased release of PGE2 facilitates tumor immune evasion. Abbreviations used: IFN-γ, interferon-γ; INFR, interferon-receptor; SLC7A11, solute carrier family 7 member 11; SLC3A2, solute carrier family 3 member 2; STAT1, signal transducer and activator of transcription 1; L-ROS, lipid reactive oxygen species; GSH, glutathione; GPX4, glutathione peroxidase 4; DAMPs, damage-associated molecular patterns; HMGB1, high mobility group box 1; CRT, calreticulin; AA, arachidonic acid; PGE2, prostaglandin E2.

**Figure 3 cells-10-00303-f003:**
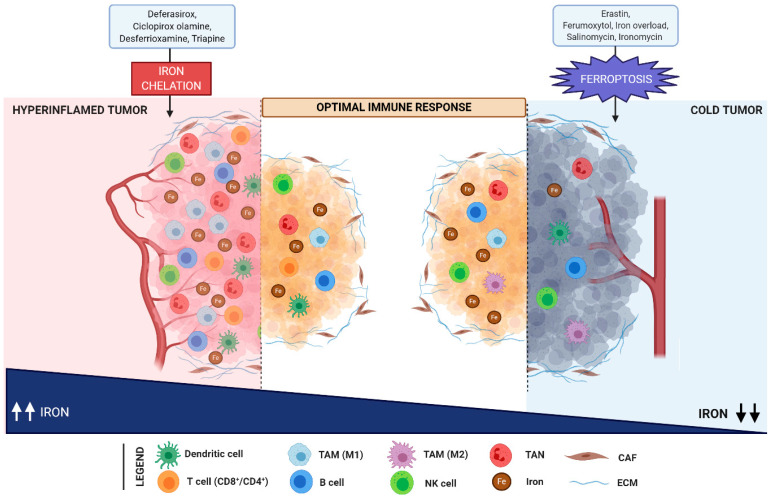
Iron levels modulate the immune response in the TME. Hyperinflamed tumors are enriched in immune cells infiltration (including dendritic cells, T- and B-cells, NK cells, TAN, and M1 macrophages) and are characterized by a high iron content. In contrast, cold tumors are poor in immune cells and are distinguished by iron deficiency. In hyperinflamed tumors, the reduction of iron overload upon treatment with iron chelators (such as deferasirox, ciclopirox olamine, desferrioxamine, and triapine), induces an efficient anti-tumor immune response. In parallel, ferroptosis mediated by erastin, ferumoxytol, salinomycin, and its synthetic derivative ironomycin foster anti-tumor immunity in cold tumors. Abbreviations used: TAM, tumor associated macrophages; TAN, tumor associated neutrophils; CAF, cancer associated fibroblasts; ECM, extracellular matrix; NK, natural killer.

**Table 1 cells-10-00303-t001:** The main proteins involved in the intracellular iron metabolism. The proteins have been divided according to the iron-related process they are involved in. Their specific function and cellular localization were subsequently reported in the last two columns.

Process	Protein	Function	Location
Cellular iron uptake	DMT1	Iron transporter of Fe^2+^	Endosome > cytosol
	DMT1	Iron transporter of Fe^2+^	Cell surface > cytosol
	Low pH	Release of Fe^3+^ from TF-Fe^3+^ (TFR1 recycled to surface)	Endosome
	STEAP2/3/4	Ferrireductase (reduces Fe^3+^ to Fe^2+^)	Endosome
	TFR1	Binds and endocytoses TF-Fe^3+^	Cell surface
	ZIP8/14	Binds and uptakes NTBI into cell	Cell surface > cytosol
	CD44	Endocytosis of iron-bound hyaluronate	Cell surface > cytosol
Cellular iron storage/release	FtH	Components of “ferritin cage”	Cytosol/mitochondria
	FtL	Components of “ferritin cage”	Cytosol
	FtMt	Mitochondrial iron storage	Mitochondria
	NCOA4	Ferritinophagy	Cytosol
Iron cellular export	CP	Ferroxidase (oxidizes Fe^2+^ to Fe^3+^)	Outer cell surface
	FPN	Fe^2+^ exporter from the cell	Cytosol > circulation
Cellular iron chaperone	PCBP1/2	Deliver iron to ferritin	Cytosol

Abbreviations used: DMT1, divalent Metal (Ion) transporter 1; STEAP2/3/4, six-transmembrane epithelial antigen of prostate 2/3/4; TFR-1, transferrin receptor; ZIP8/14, ZRT/IRT-like protein 8/14; CD44, Cell Surface Glycoprotein CD44; FtH, ferritin heavy chain 1; FtL, ferritin light chain; FtMt, mitochondrial ferritin; NCOA4, nuclear receptor coactivator 4; CP, ceruloplasmin; FPN, ferroportin; PCBP1/2, poly(C)-binding proteins 1/2.

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
