# Peer review of "Iron Metabolism in the Tumor Microenvironment—Implications for Anti-Cancer Immune Response"

_cells, 2021, doi:10.3390/cells10020303_

Round 1

Reviewer 1 Report

In this review by Sacco et al. entitled “Iron Metabolism in the Tumor Microenvironment-Implications for Anti-Cancer Immune Response”, the authors provide detailed review on the implication of iron in tumor cells and cells of the microenvironment. Iron metabolism in cancer progression and ferroptosis are emerging areas that have drawn considerable attention in the cancer community in recent years.This review article is well documented and compelling. I suggest the following points to be considered prior to publication.

1) The authors state that the M1/M2 classification of macrophages is an oversimplification. Could the authors elaborate on this comment and provide more details? Some papers the authors may wish to cite in this context:  (Nature Reviews Immunology,2011, 11, 750, Scientific Reports  2020, 10, 16554, F1000Prime Rep. 2014, 6, 13). In paragraph 3 the authors revert to the simplistic M1/M2 classification. A more general and critical discussion would be appreciated.

2) The authors state that “Compared to normal cells, tumor cells exhibit iron addiction”. In this context, recent work showed that cancer stem cells (or persister cancer cells) are addicted to iron (Mai et al. Nature Chemistry, 2017, 9, 1025–1033, Muller et al, Nature Chemistry, 2020, 12, 929). The authors should discuss this further. In line with this, salinomycin and its synthetic derivative ironomycin should be featured in Figure 3 as small molecules that can trigger ferroptosis in tumors (blue part of the figure). Note that while these molecules target lysosomal iron, they do not operate as regular tight chelators, and can promote the production of ROS and ferroptosis by sequestering iron in lysosome. The authors could also emphasize that lysosomal iron is a druggable target.

3) There is a recent study showing that iron endocytosis in cancer cells is not solely reliant on transferrin receptor 1 (TFR1) but predominantly rely on CD44 in the mesenchymal state (persister). See Muller et al, Nature Chemistry, 2020, 12, 929). This work should be cited and discussed. CD44 as an iron uptake mechanism should be added to table 1.

Through EMT, mesenchymal cancer cells (persister state, CSC) upregulate CD44/hyaluronate-mediated iron endocytosis leading to incerase iron uptake. Iron is used in mitochondrai (Krebs cycle enzyme) but also as metal catalysts of iron and aKG-dependent demethylases to deplete methyl of histone marks and unlock the expression mesenchymal genes. By doing so, this state of cells is addicted to iron, lysosomal iron increases making this state of cells vulnerable to salinomycin and ferroptosis.

4) The authors state “Iron is able to promote tumor cell growth by acting as a cofactor of enzymes”. Could the authors elaborate on this? Iron can indeed be a cofactor, but it can also act as a metal catalyst (for instance in histone demethylases see Nature Chemistry, 2020, 12, 929) and this should be stated and corrected. Iron is a master regulator of epigenetic plasticity underlying EMT and cancer progression.

5) Table 1 is incomplete. For instance PCBP1 and 2 are iron chaperones that deliver iron to ferritin (Science, 2008, 320, 1207) and should be mentioned and discussed. Mitochondrial Ferritin is also missing from this list, as well as CD44 as mentioned above. There is also a typo in the heading (rocess instead of Process).

6) The authors talk about reduction of ferric iron by STEAP3. However, there are 4 different STEAP proteins in humans. This could be discussed further (Blood, 2006, 108, 1388, J. Endocrinol. 2017, 234, R123).

7) Iron release from ferritin is not only achieved through ferritinophagy, but also by direct reduction of the iron core of intact ferritin cages (through H2O2, flavins etc.). The authors should discuss this type of iron mobilization from ferritin.

8) The authors describe the gene SLC7A11 as a “ferroptosis marker”. This reviewer finds this terminology misleading. Firstly, there is a confusion between the gene and the protein it encodes for. Secondly, talking about the system xc- as a ferroptosis marker is not correct, but rather it has been implicated in some reports that attempted to characterize ferroptotic cell death (please not that in the manuscript the authors use Xc- or xc-. Consistency would be appreciated). As a ferroptosis review, I would suggest to cited the following article: Progress in Understanding Ferroptosis and Challenges in Its Targeting for Therapeutic Benefit
Yilong Zou and Stuart L. Schreiber. Cell Chem Biol 2020).

9. The authors omit a discussion of cancer associated fibroblasts (CAFs) and iron in the tumor microenvironment. For example Oncogene, 2018, 37, 29, 4013.

A very interesting review article otherwise.

Author Response

REVIEWER 1

We really thank the reviewer for comments and suggestions that allow us to deeply strengthen some points. Please find our response in the following point-by-point letter.

1) The authors state that the M1/M2 classification of macrophages is an oversimplification. Could the authors elaborate on this comment and provide more details? Some papers the authors may wish to cite in this context: (Nature Reviews Immunology,2011, 11, 750, Scientific Reports  2020, 10, 16554, F1000Prime Rep. 2014, 6, 13). In paragraph 3 the authors revert to the simplistic M1/M2 classification. A more general and critical discussion would be appreciated.

We thank the reviewer for this comment. A deeper discussion about this point has been provided in the Introduction section. See lines 57-69 of the revised version.

2) The authors state that “Compared to normal cells, tumor cells exhibit iron addiction”. In this context, recent work showed that cancer stem cells (or persister cancer cells) are addicted to iron (Mai et al. Nature Chemistry, 2017, 9, 1025–1033, Muller et al, Nature Chemistry, 2020, 12, 929). The authors should discuss this further. In line with this, salinomycin and its synthetic derivative ironomycin should be featured in Figure 3 as small molecules that can trigger ferroptosis in tumors (blue part of the figure). Note that while these molecules target lysosomal iron, they do not operate as regular tight chelators, and can promote the production of ROS and ferroptosis by sequestering iron in lysosome. The authors could also emphasize that lysosomal iron is a druggable target.

Following the reviewer comment, further details about the role of iron in defining CSCs features have been discussed within the text (see lines 86-89; 92-98; 361-367 of the revised version) and the relative references have been added within the reference list. The potential use of salinomycin and its synthetic derivative ironomycin for triggering ferroptosis has been discussed in section 6. Salinomycin and ironomycin have been also added in Figure 3 and in the relative Figure legend.

3) There is a recent study showing that iron endocytosis in cancer cells is not solely reliant on transferrin receptor 1 (TFR1) but predominantly rely on CD44 in the mesenchymal state (persister). See Muller et al, Nature Chemistry, 2020, 12, 929). This work should be cited and discussed. CD44 as an iron uptake mechanism should be added to table 1.

Through EMT, mesenchymal cancer cells (persister state, CSC) upregulate CD44/hyaluronate-mediated iron endocytosis leading to incerase iron uptake. Iron is used in mitochondrai (Krebs cycle enzyme) but also as metal catalysts of iron and aKG-dependent demethylases to deplete methyl of histone marks and unlock the expression mesenchymal genes. By doing so, this state of cells is addicted to iron, lysosomal iron increases making this state of cells vulnerable to salinomycin and ferroptosis.

The role of CD44 as alternative iron endocytosis carrier in mesenchymal cancer cells has been discussed in section 2 (see lines 116-118 of the revised version) and the relative reference has been added within the reference list. CD44 has been added within Table 1 and within the relative caption.

4) The authors state “Iron is able to promote tumor cell growth by acting as a cofactor of enzymes”. Could the authors elaborate on this? Iron can indeed be a cofactor, but it can also act as a metal catalyst (for instance in histone demethylases see Nature Chemistry, 2020, 12, 929) and this should be stated and corrected. Iron is a master regulator of epigenetic plasticity underlying EMT and cancer progression.

The role of iron as cofactor or catalyst of enzymes has been discussed in section 1 (see lines 92-98 of the revised version) and the relative reference has been added within the reference list.

5) Table 1 is incomplete. For instance PCBP1 and 2 are iron chaperones that deliver iron to ferritin (Science, 2008, 320, 1207) and should be mentioned and discussed. Mitochondrial Ferritin is also missing from this list, as well as CD44 as mentioned above. There is also a typo in the heading (rocess instead of Process).

We thank the reviewer for noticing this missing information. PCBP1/2, FtMt (mitochondrial ferritin) and CD44 have been added within Table 1 and further discussed within the text (see lines 129-131 of the revised version).

“rocess” has been corrected.

The suggested reference has been added within the reference list.

6) The authors talk about reduction of ferric iron by STEAP3. However, there are 4 different STEAP proteins in humans. This could be discussed further (Blood, 2006, 108, 1388, J. Endocrinol. 2017, 234, R123).

This information has been corrected (see lines 113-114 of the revised version).

7) Iron release from ferritin is not only achieved through ferritinophagy, but also by direct reduction of the iron core of intact ferritin cages (through H2O2, flavins etc.). The authors should discuss this type of iron mobilization from ferritin.

This information has been added within the section 2 (see lines 122-124 of the revised version).

8) The authors describe the gene SLC7A11 as a “ferroptosis marker”. This reviewer finds this terminology misleading. Firstly, there is a confusion between the gene and the protein it encodes for. Secondly, talking about the system xc- as a ferroptosis marker is not correct, but rather it has been implicated in some reports that attempted to characterize ferroptotic cell death (please not that in the manuscript the authors use Xc- or xc-. Consistency would be appreciated). As a ferroptosis review, I would suggest to cited the following article: Progress in Understanding Ferroptosis and Challenges in Its Targeting for Therapeutic Benefit
Yilong Zou and Stuart L. Schreiber. Cell Chem Biol 2020).

We totally agree with the reviewer that the sentence he refers to is misleading. Ferroptosis marker has been correctly removed. Furthermore, we have specified that activated CD8 cells inhibit SLC7A11 gene expression (see lines 269-271 of the revised version)

“System Xc-“ is now used along the text.

As suggested by the reviewer, the recent review by Yilong Zou and Stuart L. Schreiber has been cited.

9. The authors omit a discussion of cancer associated fibroblasts (CAFs) and iron in the tumor microenvironment. For example Oncogene, 2018, 37, 29, 4013.

A discussion about the crasstalk between iron and CAFs is now reported in the Introduction section (see lines 76-82 of the revised version).

Reviewer 2 Report

The review entitled "Iron Metabolism in the Tumor Microenvironment—Implications for Anti Cancer Immune Response" by A. Sacco et al. Is an interesting review regarding how the bioavailability of this metal shapes complex metabolic programs within the tumor microenvironment and how this affects both tumor-associated macrophages and tumor-infiltrating lymphocytes functions. The subject is new and promising. The introduction and discussions are well and clearly write. The references were well selected and up-dated. From my point of view is suitable for publication.

Author Response

We really thank the reviewer for the positive evaluation.

Reviewer 3 Report

The authors propose a review on iron metabolism in the tumor microenvironment and the implications for anti-cancer immune response.

After an introduction on iron homeostasis, the authors discuss iron handling by Tumor-associated macrophages and lymphocytes in the context of carcinogenesis. A section is next devoted to ferroptosis. Finally, iron is discussed as a potential target in the context of hyperinflammed tumors and cold tumors.

This is an interesting review based on recent data of the literature, (including articles published by the authors of the review themselves), and well illustrated.

This review is timely as there was some recent significant advance in this field, which required putting into context.

However, overall, some sentences need to be rewritten 1) to increase clarity or 2) to avoid plagiarism.

Major points :

I found, by chance, sentences copied « word to word » from a recent published review, such as :

-  Lines 276 page 7 : « Subsequently, the same team reported that IFN-γ derived from immunotherapy-activated CD8+ T cells synergizes with radiotherapy-activated ataxia-telangiectasia mutated (ATM) to induce ferroptosis in human fibrosarcoma cells and melanoma cells  [107]. »

Cf. Tang et al. Ferroptosis, necroptosis, and pyroptosis in anticancer immunity. J. Hematol. Oncol. 2020 : « Subsequently, the same team reported that IFN-γ derived from immunotherapy-activated CD8+ T cells synergizes with radiotherapy-activated ataxia-telangiectasia mutated (ATM) to induce ferroptosis in human fibrosarcoma cells and melanoma cells [32] ».

OR

« HMGB1 is a non-histone chromatin-binding protein localized in the nucleus, where it interacts with DNA and regulates transcription [110–112]. » «  When released from dying cells, HMBG1 exerts potent immunostimulatory effects by interacting with distinct PRRs (TLR2, TLR4, and RAGE) [113], and thus it is required for the immunogenicity of cancer cells [114]. During chemotherapy- or radiotherapy-induced cell death, HMGB1 is released from dying cells and signals through TLR4-MyD88 axis on DCs, facilitating antigen processing.. »

from Serrano-del Valle, A.; Anel, A.; Naval, J.; Marzo, I. Immunogenic cell death and immunotherapy of multiple myeloma. Front. 636 Cell Dev. Biol. 2019.

Check the review for other plagiarisms.

Minor points :

- Line 121 page 3 :

« Hepcidin (…) binds to the extracellular part of FPN targeting it for degradation. »

This is not the only mechanism. A recent article published in Nature (Billesbølle et L., Nature, 2020) showed that hepcidin binds FPN in a central cavity between the N and C domains, acting as a molecular cork to completely occlude the iron efflux pathway. Moreover, this article suggests a model for hepcidin regulation of ferroportin, in which only ferroportin molecules loaded with iron are targeted for degradation.

- Lines160-163 page 4. The following sentence is not clear to me. Could the authors please clarify ?

« Once released, heme is then degraded via Heme Oxygenase (HMOX-1), the inducible isoform of the heme-degrading enzyme which induces the downregulation of the transcription factor Bach-1 [72], thereby allowing the transcription of FPN, and at the same time, is critical for driving resolution of inflammatory responses [67]. »

- Lines 229-231 pages 6. The sentences « FT secreted by TAMS … are reported as well”, should be in the section on TAMs and not in the section on T cell function.

- Lines 279-281 page 7 The following sentence is confused : « In addition, analysis of gene expression revealed that melanoma patients showing a clinical benefit of immunotherapy expressed gene signatures suggesting active T cell-induced ferroptosis. »

- There are a lot references to reviews. In some cases, references to original articles could be useful.

- Typo legend Figure1 (line 203) : can secretE

Author Response

We really thank the reviewer for comments and suggestions that allow us to deeply strengthen some points. Please find our response in the following point-by-point letter.

The authors propose a review on iron metabolism in the tumor microenvironment and the implications for anti-cancer immune response. After an introduction on iron homeostasis, the authors discuss iron handling by Tumor-associated macrophages and lymphocytes in the context of carcinogenesis. A section is next devoted to ferroptosis. Finally, iron is discussed as a potential target in the context of hyperinflammed tumors and cold tumors.This is an interesting review based on recent data of the literature, (including articles published by the authors of the review themselves), and well illustrated. This review is timely as there was some recent significant advance in this field, which required putting into context. However, overall, some sentences need to be rewritten 1) to increase clarity or 2) to avoid plagiarism.

We thank the reviewer for this comment, and we apologize for this inconvenience. All the manuscript has been checked for other plagiarisms and eventually corrected.

Major points :

I found, by chance, sentences copied « word to word » from a recent published review, such as :

-  Lines 276 page 7 : « Subsequently, the same team reported that IFN-γ derived from immunotherapy-activated CD8+ T cells synergizes with radiotherapy-activated ataxia-telangiectasia mutated (ATM) to induce ferroptosis in human fibrosarcoma cells and melanoma cells  [107]. »

Cf. Tang et al. Ferroptosis, necroptosis, and pyroptosis in anticancer immunity. J. Hematol. Oncol. 2020 : « Subsequently, the same team reported that IFN-γ derived from immunotherapy-activated CD8+ T cells synergizes with radiotherapy-activated ataxia-telangiectasia mutated (ATM) to induce ferroptosis in human fibrosarcoma cells and melanoma cells [32] ».

OR

« HMGB1 is a non-histone chromatin-binding protein localized in the nucleus, where it interacts with DNA and regulates transcription [110–112]. » « When released from dying cells, HMBG1 exerts potent immunostimulatory effects by interacting with distinct PRRs (TLR2, TLR4, and RAGE) [113], and thus it is required for the immunogenicity of cancer cells [114]. During chemotherapy- or radiotherapy-induced cell death, HMGB1 is released from dying cells and signals through TLR4-MyD88 axis on DCs, facilitating antigen processing.. » from Serrano-del Valle, A.; Anel, A.; Naval, J.; Marzo, I. Immunogenic cell death and immunotherapy of multiple myeloma. Front. 636 Cell Dev. Biol. 2019.

Check the review for other plagiarisms.

We thank the reviewer for this comment, and we apologize for this inconvenience. All the manuscript has been checked for other plagiarisms and eventually corrected.

Minor points :

- Line 121 page 3 :

« Hepcidin (…) binds to the extracellular part of FPN targeting it for degradation. »

This is not the only mechanism. A recent article published in Nature (Billesbølle et L., Nature, 2020) showed that hepcidin binds FPN in a central cavity between the N and C domains, acting as a molecular cork to completely occlude the iron efflux pathway. Moreover, this article suggests a model for hepcidin regulation of ferroportin, in which only ferroportin molecules loaded with iron are targeted for degradation.

We thank the reviewer for this suggestion. This hepcidin-mediated FPN regulation mechanisms are discussed in section 2 (see lines 140-143 of the revised version).

- Lines160-163 page 4. The following sentence is not clear to me. Could the authors please clarify ?

« Once released, heme is then degraded via Heme Oxygenase (HMOX-1), the inducible isoform of the heme-degrading enzyme which induces the downregulation of the transcription factor Bach-1 [72], thereby allowing the transcription of FPN, and at the same time, is critical for driving resolution of inflammatory responses [67]. »

We agree with the reviewer that this sentence can be misleading, thus it has been rewritten (See lines 184-188 of the revised version)

- Lines 229-231 pages 6. The sentences « FT secreted by TAMS … are reported as well”, should be in the section on TAMs and not in the section on T cell function.

The sentences « FT secreted by TAMS … are reported as well” have been correctly moved in the TAMs section (see lines 196-198 of the revised version).

- Lines 279-281 page 7 The following sentence is confused : « In addition, analysis of gene expression revealed that melanoma patients showing a clinical benefit of immunotherapy expressed gene signatures suggesting active T cell-induced ferroptosis. »

The manuscript by Wang et al highlighted that immunotherapy-activated CD8+ T cells promote ferroptosis in cancer cells by repressing the transcription of SLC3A2 and SLC7A11 genes, encoding for the two subunits of system Xc-.  In agreement, the same team found that in melanoma patients which benefit from nivolumab therapy, a higher T cell infiltration as well as an increased release of IFN-gamma correlate with low expression of SLC3A2 gene in the relative cancer cells. See lines 268-280 of the revised version.

- There are a lot references to reviews. In some cases, references to original articles could be useful.

Several references to reviews (46 out of 65) have been replaced with references to original articles.

- Typo legend Figure1 (line 203) : can secretE

The typing error has been corrected.

Reviewer 4 Report

This is a comprehensive and balanced review on iron in the context of tumor microenvironment.  It discusses recent advances and controversies in this field and links them to possible therapeutic interventions. There are some issues that require attention, especially in the introductory paragraphs. Details are provided below:

  1. Lines 90-91; The sentence states that NTBI enters cells via ZIP8/14 following reduction by PRNP. However, this is oversimplified. There could be more mechanisms for NTBI uptake (for instance involving DMT1); Zip14 is essential for hepatic iron overload in hemochromatosis. The role of PRNP in NTBI uptake does not appear to be firmly established (there is only one relevant primary paper, which is not cited here). PRNP is (correctly) not mentioned in Table 1.

  1. Lines 94-95; It is implied that ferritin prevents ROS formation by acting as ferroxidase. The correct phrasing is that ferritin prevents ROS formation by sequestering Fe2+, which is coupled to oxidation to Fe3+.

  1. Lines 107-108; The sentence implies that frataxin only operates in the heme biosynthetic pathway. However, frataxin also plays an essential role in Fe-S cluster biogenesis.

  1. Lines 110-111; “When intracellular iron levels exceed the storing potential, any excess iron is exported from the cell”. This is not true for iron-exporting cells, such as erythrophagocytic macrophages.

  1. Table 1; There is a typo (p is missing from the word process). The function of HEPH and HEPHL1 is not indicated. Apart from this, I am not sure whether HEPHL1 should be listed on this Table since evidence for an important in vivo function is missing.

  1. Figure 1 discusses a potential role of DMT1 as NTBI transporter, but this is somehow ignored in the text (see also comment 1).

  1. In the legend to Figure 1, please correct “metabolization” to “metabolism”.

Author Response

We really thank the reviewer for comments and suggestions that allow us to deeply strengthen some points. Please find our response in the following point-by-point letter.

  1. Lines 90-91; The sentence states that NTBI enters cells via ZIP8/14 following reduction by PRNP. However, this is oversimplified. There could be more mechanisms for NTBI uptake (for instance involving DMT1); Zip14 is essential for hepatic iron overload in hemochromatosis. The role of PRNP in NTBI uptake does not appear to be firmly established (there is only one relevant primary paper, which is not cited here). PRNP is (correctly) not mentioned in Table 1.

We thank the reviewer for this comment. We have now added DMT1, ZIP14 and L-type voltage-dependent calcium channels as molecular channels involved in NTBI intracellular uptake (see lines 114-116 of the revised version). Since the role of PRNP has not been clearly defined, we decided not to cite it within the text.

  1. Lines 94-95; It is implied that ferritin prevents ROS formation by acting as ferroxidase. The correct phrasing is that ferritin prevents ROS formation by sequestering Fe2+, which is coupled to oxidation to Fe3+.

Following the reviewers’ suggestion “ferritin prevents ROS formation by sequestering Fe2+, which is coupled to oxidation to Fe3+” sentence is now used in lines 121-122 of the revised version.

  1. Lines 107-108; The sentence implies that frataxin only operates in the heme biosynthetic pathway. However, frataxin also plays an essential role in Fe-S cluster biogenesis.

The indicated sentence has been corrected to specify the role of frataxin in Fe-S cluster biogenesis as well.

See lines 132-134 of the revised version.

  1. Lines 110-111; “When intracellular iron levels exceed the storing potential, any excess iron is exported from the cell”. This is not true for iron-exporting cells, such as erythrophagocytic macrophages.

Following the reviewers’ suggestion this sentence has been corrected to avoid any misreading.

  1. Table 1; There is a typo (p is missing from the word process). The function of HEPH and HEPHL1 is not indicated. Apart from this, I am not sure whether HEPHL1 should be listed on this Table since evidence for an important in vivo function is missing.

The typing error has been corrected.

According to the reviewers’ suggestion HEPH and HEPHL1 have been removed from Table 1 and the relative caption.

  1. Figure 1 discusses a potential role of DMT1 as NTBI transporter, but this is somehow ignored in the text (see also comment 1).

Upon comment 1, we have now reported within the text as well as within Figure 1 the membranes carriers/channels (DMT1, ZIP8, ZIP14 and L-type voltage-dependent calcium channels) involved in NTBI iron uptake.

See lines 114-116 of the revised version.

  1. In the legend to Figure 1, please correct “metabolization” to “metabolism”.

Metabolization has been replaced with metabolism.